# Multicellular Modelling of Difficult-to-Treat Gastrointestinal Cancers: Current Possibilities and Challenges

**DOI:** 10.3390/ijms23063147

**Published:** 2022-03-15

**Authors:** Sarah K. Hakuno, Ellis Michiels, Eleonore B. Kuhlemaijer, Ilse Rooman, Lukas J. A. C. Hawinkels, Marije Slingerland

**Affiliations:** 1Department of Gastroenterology and Hepatology, Leiden University Medical Center, 2300 RC Leiden, The Netherlands; s.k.hakuno@lumc.nl (S.K.H.); e.b.kuhlemaijer@lumc.nl (E.B.K.); 2InnoSer Belgium NV, 3590 Diepenbeek, Belgium; ellis.michiels@vub.be; 3Department of Medical and Molecular Oncology, Free University Brussels, 1090 Brussels, Belgium; ilse.rooman@vub.be; 4Department of Medical Oncology, Leiden University Medical Center, 2300 RC Leiden, The Netherlands; m.slingerland@lumc.nl

**Keywords:** patient-derived models, patient-derived organoids, patient-derived xenografts, gastric cancer, pancreatic cancer, multicellular models, tumour modelling, co-cultures, humanised mice, tumour microenvironment

## Abstract

Cancers affecting the gastrointestinal system are highly prevalent and their incidence is still increasing. Among them, gastric and pancreatic cancers have a dismal prognosis (survival of 5–20%) and are defined as difficult-to-treat cancers. This reflects the urge for novel therapeutic targets and aims for personalised therapies. As a prerequisite for identifying targets and test therapeutic interventions, the development of well-established, translational and reliable preclinical research models is instrumental. This review discusses the development, advantages and limitations of both patient-derived organoids (PDO) and patient-derived xenografts (PDX) for gastric and pancreatic ductal adenocarcinoma (PDAC). First and next generation multicellular PDO/PDX models are believed to faithfully generate a patient-specific avatar in a preclinical setting, opening novel therapeutic directions for these difficult-to-treat cancers. Excitingly, future opportunities such as PDO co-cultures with immune or stromal cells, organoid-on-a-chip models and humanised PDXs are the basis of a completely new area, offering close-to-human models. These tools can be exploited to understand cancer heterogeneity, which is indispensable to pave the way towards more tumour-specific therapies and, with that, better survival for patients.

## 1. Introduction

The survival of cancer patients has significantly improved during the last decade as a result of a better understanding of cancer biology and development of novel promising therapies, such as immunotherapy. Despite that, cancer incidence is still increasing, and cancer is a major cause of death worldwide. However, unfortunately, some cancer types still show high relapse and mortality rates. Gastric and pancreatic cancers, two types of gastrointestinal cancer, are known to have limited treatment options, progress quickly and result in high mortality [1]. This emphasises the need for novel and effective therapeutic interventions. Previous cancer models have mainly relied on monolayer in vitro cell cultures and subcutaneous cellular grafting in immunodeficient mice. During the last decade, major advances have been made with the development of patient-derived organoid (PDO) cultures and engraftment of freshly isolated human tumour tissues in mice (patient-derived xenografts, PDXs). While differences exist regarding costs, scalability and the contribution of the tumour microenvironment (TME), these models benefit from a high degree of biological concordance and stability over time. Still, a lot of work needs to be done to completely recapitulate all elements of a human tumour in preclinical research. Here, we will discuss advantages and limitations of first generation PDO/PDX models in the context of two difficult-to-treat gastrointestinal cancers, with particular focus on more recent efforts to create three-dimensional (3D) multicellular models starting from PDOs and the development of humanised PDX mice.

## 2. Difficult-to-Treat Gastrointestinal Cancers

Gastric cancer and pancreatic cancer are some of the most lethal cancers known, with both occurring at a relatively high incidence. The dismal prognosis is partly due to late diagnosis and the limited treatment options currently available. In both of these cancers, neoplastic cells are surrounded by a significant amount of non-epithelial/stromal cells, such as cancer-associated fibroblasts (CAFs), infiltrating immune cells and endothelial cells forming the tumour blood vessels, all in a dense extracellular matrix. This TME favours tumour growth and progression and it functions as a physiological barrier, hampering oxygen supply and the delivery of therapeutics.

### 2.1. Gastric Cancer

There are 1,000,000 new cases of gastric cancer (GC) annually, a substantial amount of which occur in eastern Asia and Europe. It ranks fifth in terms of incidence and fourth in terms of mortality globally [1]. GC is known to have poor prognosis; 5-year survival rates of only 19–31% have been reported, resulting in approximately 769,000 deaths worldwide in 2020 [1,2], partly due to late diagnosis [3]. GC metastasizes mainly to the liver, peritoneum, lungs, bones, and lymph system [4,5,6]. GC is a heterogeneous disease, hampering the development of effective treatments [7,8]. Histologically, tumours can be classified based on the Lauren or the more elaborate World Health Organisation (WHO) classification [9,10]. In addition to these classifications, GCs can be characterised on the molecular level [11]. Research has reported mutations in *KRAS*, *APC*, *ARID1A*, *PIK3CA*, *ERBB3*, *HLA-B*, *SMAD4*, *STK11*, *PTEN*, and *BMPR1A*, amongst others, with the most frequent reports of mutations in *CDH1* and *TP53* [11,12,13,14,15]. The TME also plays a role in tumour progression, as it has been shown that a high amount of tumour stroma is associated with worse five-year survival in GC [16]. Curative treatment for early-stage cancer involves surgical or endoscopic tumour resection, often in combination with (neo)adjuvant chemotherapy [17,18]. Treatment selection is currently based on the tumour stage. For most patients diagnosed with advanced (unresectable or metastatic) disease, palliative treatment and/or best supportive care are the only available options [17]. The use of biologicals is evolving, with anti-human epidermal growth factor receptor 2 (HER2) therapy promising in cancers with overexpression of HER2 in first line treatment and ramucirumab in second line treatment [19,20,21]. Furthermore, immuno-oncology with checkpoint inhibition and immune stimulation has evolved in the field of GC. Preliminary data show a significant survival benefit in GC patients treated with immunochemotherapy [22].

### 2.2. Pancreatic Ductal Adenocarcinoma

According to recent global cancer statistics, almost half a million people are diagnosed worldwide with pancreatic cancer every year and numbers keep rising [1]. Pancreatic ductal adenocarcinoma (PDAC) is the most prevalent type of pancreatic cancer and is known for a very poor prognosis and high mortality rates. It ranks seventh in terms of mortality worldwide and the 5-year survival rate is <10%. Similar to GC, PDAC is known to be a heterogeneous disease, which has therapeutic implications. As with other cancers, the TNM classification system is used to define different PDAC stages. Histologically, PDAC is characterised by atypical tumour glands surrounded by a prominent desmoplastic stromal mass. Genomic analyses have identified *KRAS*, *TP53*, *CDKN2A*, and *SMAD4* to be the most commonly occurring genetic alterations, with *KRAS* having a 90% prevalence. On a molecular level, transcriptomic analysis has revealed that PDAC can be divided into two major groups, the “classical” and “basal-like” subtypes, with the latter having the worst prognosis [23,24,25,26,27]. PDAC usually metastasizes to the liver, peritoneum, and to the lungs [28]. Only 15–20% of PDAC cases are suitable for surgical resection and the majority of patients usually relapse within 5 years post-surgery [29]. The standard of care for (non)-resectable patients includes (neo)-adjuvant chemotherapy regimens, but this has limited efficiency, with prolonged survival times of 6–18 months [29]. Evidence is emerging that molecular subtypes and/or the presence of certain mutations, such as *KRAS^G12C^*, might be of clinical importance in the future to improve therapeutic responses to specific inhibitors [30]. To summarise, PDAC harbours a complex mutational landscape and malignant cells are protected by a dense stromal barrier, explaining the difficulty of successful treatment.

## 3. First Generation PDO/PDX Modelling

Historically, two-dimensionally (2D) grown cell lines derived from human tumour tissues have been used for cancer research for many years. Being relatively simple, stable, immortal, and inexpensive, cell-line based studies have shown significant value and have led to insights in oncologic signaling and the identification of novel therapeutic targets [31]. Regardless, extrapolating preclinical findings to the clinic has become a major challenge in drug discovery because 2D models only partly represent the in vivo situation. Not all cancer cell subclones are reflected accurately in a (clonal) cell line and, importantly, there is a lack of the TME, resulting in no 3D cell–cell interactions and/or tumour–stroma interactions [31,32]. Culture conditions are also known to propagate phenotypic features that are not necessarily representative of the in vivo situation [33]. Moving forward, inclusion of patient-derived 3D models has significantly improved the representation of the in vivo situation. Here, we review PDO and PDX to better model gastric and pancreatic cancer.

### 3.1. Patient-Derived Organoids (PDOs)

A major breakthrough in oncology research was the discovery of how to establish human 3D organoids from human tissues and cancers [34]. Long-term murine and human organoid cultures have been generated from stem cells by stimulation of the Wnt pathway and inhibition of the Transforming Growth Factor (TGF)-β pathway [35]. Cells obtained from surgical resections and biopsies of human tumours have been successfully used to establish PDOs with high representativeness of the original tumour on a morphological, genetic, and phenotypic level [36,37,38]. Due to the method of culturing, PDOs have certain advantages over other cancer models (Table 1). Compared to in vivo experiments, PDOs can be generated in small volumes and at low costs. It normally takes two weeks to one month from tissue digestion until a stable PDO culture is established [38]. Culture conditions make PDOs fit for transcriptome and high-throughput drug screening, allowing in-depth studies in development and diseases, but also offering novel possibilities for more personalised treatment [39,40]. As an important drawback, the success rate of culturing PDO is highly dependent on the tumour origin, cellularity, tissue size, amount of stroma, and whether the patient received neoadjuvant therapy or not [41]. This demands stan-darisation of protocols to create highly representative living biobanks for these and other cancer indications.

#### 3.1.1. Gastric Cancer PDOs

The Clevers group at the Hubrecht Institute was the first to report the establishment of gastric organoids, in 2010. They cultured murine leucine-rich repeat-containing G-protein coupled receptor 5 (Lgr5)^+^ gastric adult stem cells in Matrigel overlaid with standard PDO growth medium, supplemented with Wnt3a and fibroblast growth factor 10 (FGF10) [35]. In 2015, they described the generation of PDOs from human GC material [42]. GC-PDOs have been assessed multiple times for their biological resemblance and stability for prolonged cell culture. Genetic characteristics, including copy number alterations (CNA), driver mutations, and ploidy have been demonstrated to match the original tumour tissue well [38,43,44]. Heterogeneity in HER2 and fibroblast growth factor receptor 2 (FGFR2) overexpressing cells in fresh tumour tissue was shown to be retained in PDOs [44], although differences in HER2 expression between primary tumour and PDOs have been reported as well [45]. Next to genetic stability, it is important to know if therapeutic responses are reflected in organoids. In that regard, it has been noted that PDOs respond better to treatment than patients do, probably because of the absence of the tumour stroma and enzymes involved in drug metabolism [41].

GC-PDOs have already led to the discovery of potential drug targets and to the testing of potential therapeutics. MicroRNAs have been implicated in the development of cancer [46], but their role in GC development has not been confirmed yet. Two independent studies illustrated a crucial role for two specific microRNAs (miR-324-3p and miR-1265) acting as an oncogene or tumour suppressor gene in GC-PDOs [47,48]. Another study described the drug efficiency of Ethalesen, a thioredoxin reductase inhibitor, which significantly reduced GC-PDO growth [49]. The applicability of GC-PDOs for high-throughput drug screening was demonstrated in a study that tested 37 anti-cancer compounds in nine PDOs. Screening took less than two weeks per organoid, showing the feasibility of achieving reproducible results [44]. An ongoing clinical trial is comparing responses to standard of care chemotherapeutics between patients and PDOs [50]. PDOs are established from biopsies taken before and after treatment to investigate the correlation between the in vitro response and the in vivo response of patients. Evaluation is performed with both histological and omics-based analysis. With an estimated forty participants, this study will contribute to further assessing the representativeness of PDOs from GC patients.

#### 3.1.2. PDAC PDOs

In 2015, the Clevers and Tuveson research groups were the first to establish murine- and human-derived normal and pancreatic cancer organoids [36,51]. PDOs were grown in the presence of Matrigel cultured in an intestinal growth medium, supplemented with R-SPONDIN1, a Wnt pathway modulator. Engraftment of PDAC-PDOs into immunodeficient mice resulted in the formation of neoplastic lesions that progressed into invasive and metastatic PDAC [51]. Typical PDAC driver mutations, such as *KRAS* and *SMAD4*, are maintained in PDO models, as was shown by targeted sequencing [51]. Histological and genomic stability compared to the original tumours was later confirmed in other studies as well [52,53,54]. As PDAC-PDOs only represent the epithelial tumour cells, this model has been extensively exploited to study tumour biology, transcriptomics, and drug screening [52,55]. The latter study demonstrated that therapeutic responses in PDOs reliably correlated with clinical responses. This strategy allows patients to be stratified into ‘responders’ versus ‘non-responders’ to specific drugs, opening opportunities for personalised treatment. Additionally this study defined a transcriptomic chemosensitivity signature [52]. However, given the large amount of stroma observed in PDAC, the clinical translation might be challenging, and this is an important limitation of current PDO-based (therapeutic) studies.

### 3.2. Patient-Derived Xenografts (PDXs)

One of the disadvantages of PDOs is that they do not fully represent the clinical situation, mainly due to the lack of the tumour stroma/TME and blood flow (Table 1). To be able to evaluate this, PDX studies have been performed in which a human tumour is grown in an immunodeficient mouse. Primary (tumour) tissue, obtained via surgical resection or biopsy [56,57], is engrafted heterotopically (often subcutaneously) or orthotopically in the organ of tumour origin. Subcutaneous implantation is a relatively simple, less invasive method and growth can easily be monitored in vivo. However, heterotopic PDX tumours represent the former tumour environment less accurately and do not facilitate invasion and metastasis to a similar extent, compared to orthotopic transplantation [58]. As a drawback, orthotopic PDX is often more difficult to monitor in vivo [59]. Once PDX engraftment and growth is established, the tissue can be passed to new mice, expanding the amount of tumour tissue for subsequent analysis [60]. Several researchers describe a phenomenon of serial passages having a shorter latency, which increases the tumour take rates [56,61]. This might be related to the formation of the required mouse vasculature in the PDX tissue after the first passage [62], thereby accelerating subsequent tumour growth. Different PDX generations show a 95–100% concordance for histology and differentiation status. Research also shows high concordance of key driver mutations in PDXs and their corresponding human primary tumours [63]. Retention of the original tumour characteristics makes PDX a more accurate model to predict therapeutic responses, compared to traditional cell-line based or non-human based animal models.

Apart from the advantages, there are some challenges associated with PDX models (Table 1). Firstly, tumour growth after engraftment generally has a low efficiency. Factors that influence the success rate are tissue metabolism, ischemia time, sample size, surgical procedure, tumour type, tumour stage, metastatic properties, and mouse strain [64,65]. Another limitation is that, after engraftment of patient tissue into a mouse, the patient-derived stromal and immune cells do not proliferate [66], resulting in their replacement with murine stromal cells. These cells are fully functional, but this hampers the use of PDX when it comes to evaluation of stroma-targeted therapies. Furthermore, the long time (from months to over one year) it takes to generate the primary PDX model limits the potential of the model in various research and clinical applications, such as personalised medicine.

In summary, PDX models allow increased understanding of tumour growth and dynamic tumour- (mouse) stroma interactions that regulate invasiveness and distant metastasis. Moreover, the high histological resemblance provides new oncological insights and opens interesting diagnostic, prognostic, and therapeutic applications towards more personalised medicine for difficult-to-treat cancers. More experience needs to be gained to overcome the current limitations of PDX development in the context of GC and PDAC, which we will discuss further below.

#### 3.2.1. Gastric Cancer PDXs

For GC, PDX models have been firmly established. Success rates for initial implantation vary between 15.1% and 34.1% [56,60,61,64,67,68]. The parameters for successful engraftment in GC were reported to be associated to tumour subtype, tumour cell percentage and degree of necrosis [60]. Other studies have reported the influence of prior chemotherapy, patient sex, and overall procedure time on PDX success rates [56,60,68]. Molecular targets for drug therapy have been evaluated for GC-PDX, and results showed that expression of the genes *FGFR2*, *MET* and *HER2* in GC PDXs is representative of the original patient tumour [56,61,68]. Transcriptomic differences between original tumours and their derived PDX models have been linked to discrepancies in the Lauren classification and tumour heterogeneity [61,68]. Although literature generally reports high representativeness of patient tumours by PDXs, it is notable that some GC-PDXs have been found to have a different histology from the original tumour, illustrating the need for more reliable PDX models.

GC-PDX models have also been successfully used for drug discovery. Comparing the chemosensitivity of GC-PDXs to the original tumour shows that 80% of the cases exhibit similar responses [68]. Another study that investigated the effects of known cancer-related drugs on 50 GC-PDX models could match drug-mediated anti-tumour effects to specific gene expression profiles [69]. GC-PDXs were confirmed to be predictive for therapy when evaluating treatment responses to an epidermal growth factor receptor (EGFR) inhibitor (cetuximab) and HER2 inhibitor (trastuzumab), which is a standard of care for patients with advanced HER2^+^ GC [70,71]. Other reported findings include the evaluation of potential synergistic combinations of several anti-cancer drugs, with the aim of finding effective drug combinations in a personalised manner. It is suggested that co-treatment with trastuzumab and cetuximab could be the most effective therapy for EGFR^+^ and HER2^+^ GC patients [72]. However, clinical responses to this combinational therapy remain, to our knowledge, as yet unpublished.

#### 3.2.2. PDAC PDXs

Since 2006, PDAC PDXs have been widely used in preclinical research. Tumour tissue is engrafted subcutaneously or orthotopically in the pancreas. For orthotopic modelling, fresh primary tissue is usually implanted into the pancreatic head, resulting in fast tumour growth and lymph node invasion, but no formation of distant metastasis. Studies on PDAC PDX have reported an engraftment success rate between 20 and 80%, highly dependent on cellular viability, the presence of necrotic areas, and the amount of stromal cells [73,74]. Important concordance (90–97%) between the PDX model and the original tumour could be confirmed by the presence of important driver mutations of PDAC, such as *KRAS* mutations, that can be maintained for over 30 PDAC PDX generations [75,76]. Although PDX models metastasize very poorly in general, one study showed that human PDAC cells could be detected in the blood of PDAC-PDX mice, demonstrating the potential to (partially) mimic the metastatic process in PDX mice [77].

PDAC PDX models have been shown to be useful for biomarker discovery, for testing chemosensitivity, and development of novel targeted therapies [78,79,80]. Recently, subtype-specific transcriptional signatures of both tumour and stroma have been validated in PDAC PDX (and PDO) models. These results open a new window to develop subtype-specific and/or stroma-targeting molecules that could improve the survival of PDAC patients in the future [81,82].

## 4. Next Generation PDO/PDX Modelling

Irrespective of the promising PDO and PDX models that were discussed above, these models require further optimisation before broader clinical implementation becomes possible. Recent progress has been made to develop complex organotypic PDO/PDX-based models to closely mimic the complexity of a human tumour. An overview of the first and next generation patient-derived models can be seen in Figure 1.

### 4.1. Multicellular PDO Models

Despite all of their advantages compared to conventional 2D cell culture, current PDO models do not fully recapitulate the complex cellular environment of a human cancer, since they lack the TME, a crucial component when investigating tumour development, progression and treatment options. Given the increasing efficacy of immunotherapy, the development of multicellular in vitro models involving infiltrating immune cells has gained interest. Adding immune cells to the PDO models was reported for the first time in the context of colorectal cancer and non-small-cell lung cancer [83]. Hereby, peripheral T-cells were added to tumour PDO cultures, resulting in the enrichment of tumour reactive T-cells that were able to specifically kill the tumour PDO [84]. As TME also contains other important elements, such as CAFs, blood vessels and neurons, in vitro modelling is currently moving forward to complex organotypic systems with close-to-human representativity.

#### 4.1.1. PDO/Immune Cell Co-Cultures

The methodology of co-cultures of murine GC organoids with immune cells has been described previously [85]. Mouse-derived normal and/or tumour GC organoids were cultured with mouse-derived dendritic cells (DCs) and cytotoxic T-cells. It was postulated that these co-cultures could eventually be used as a preclinical prediction tool to assess the efficacy of immunotherapeutics in GC patients. Similar reports are scarce in the context of PDAC, which is known as an immune-suppressed or ‘cold’ tumour. One paper investigated the depletion of myeloid-derived suppressor cells as a potential therapeutic option for PDAC patients using immune cell/PDO co-cultures. Their results confirmed the immune-suppressive role of myeloid-derived suppressor cells, and thereby could explain the limited response of immune checkpoint inhibitors in PDAC patients in clinical trials [86]. To summarise, these data demonstrate that multicellular PDO co-cultures with immune cells are a feasible method to develop more complex research models with improved similarity to human tumours.

#### 4.1.2. PDO/CAF Co-Cultures

Given the fact that GC and PDAC can be composed of, respectively, 50% and 70% non epithelial/stromal cells [16,87], of which the most abundant cell type is the CAF, it would be a big step forward to add CAFs to conventional PDO cultures. In 2016, a method to co-culture GC organoids with immortalised gastric fibroblasts was reported [88]. In PDAC, co-cultures of both murine- and human-derived PDOs with CAFs were first reported in 2017, showing an important cellular interplay that stimulates the production of desmoplastic stroma within the cultures. Interestingly, this experimental set-up also led to the identification of two distinct CAF subpopulations (iCAF and myCAF) [89], which we have currently also observed in our recently developed multicellular PDAC mini tumours (Harryvan et al., in revision). Additional work has demonstrated the feasibility of developing multicellular co-cultures of PDOs with matched patient-derived fibroblasts and/or T-cells. The authors could observe activation in a subset of fibroblasts, and found evidence for interleukin 6 (IL6)-mediated signalling between tumour cells and stroma, contributing to their survival and growth [90].

#### 4.1.3. Organoid-on-a-Chip

To further improve PDO modelling as a bridge between in vitro and in vivo modelling, it is necessary to also add additional cell types found in the tumour microenvironment, such as immune cells, endothelial cells and pericytes, to traditional PDO culture to fully recapitulate the TME. Recent progress has been made to generate highly complex multicellular tumour models using 3D bioprinting and organ-on-a-chip platforms. This topic has been reviewed extensively elsewhere [91]. The authors conclude that the combination of organoids and organ-on-a-chip technology offer important opportunities for drug discovery, personalised medicine and regenerative medicine. With the addition of other cell types to reflect the TME more faithfully, addressing the biophysical microenvironment is also of importance. A first of its kind co-culture model made use of the “gut organoid flow chip (GOFlowChip)”, a system that allows long-term luminal flow, a critically important physiologic parameter of the human gut [92]. These authors were able to optimise the model for gastric organoids and tested the addition of DCs, resulting in a tissue chip-based co-culture model of human gastric organoids and human immune cells [93]. An important factor to consider when developing such a model is the type of extracellular matrix being used. A commercially available synthetic hydrogel (VitroGel^®^-ORGANOID-3; TheWell Biosciences, North Brunswick Township, NJ, USA) was used during these experiments as an alternative for Matrigel, leading to more reliable and reproducible results.

### 4.2. Humanised PDX Models

To overcome the current limitations of the PDX models (e.g., murine tumour stroma, no immune cells), advanced PDX models are being developed, aiming to create a more human-like animal model by introducing human immune cells into conventional PDX models. These so called “humanised” mice are generated by implanting CD34^+^ hematopoietic stem and precursor cells [94] into an immunodeficient mouse or a mouse with a human leukocyte antigen (HLA)-expressing thymus [95,96,97]. As a result, a humanised mouse will have circulating human immune cells. This innovative humanised animal model appears to be superior in terms of metastatic potential compared to immunodeficient mice [98]. This recent breakthrough provides the possibility of investigating immune infiltration, testing immuno-therapeutics and performing immune-oncology screens [66,99,100]. As a disadvantage, engrafted immune cells will often not be HLA compatible with the implanted PDX, hampering full clinical representativeness. Furthermore, they come at high costs and with demanding logistics. Still, these models are a valuable addition to current patient-derived models. Below we discuss the current studies on humanised PDX for GC and PDAC.

In general, humanised PDX models to study GC and PDAC are still limited at the moment. Probably, the establishment and use of such models will increase in the next decade and will offer significant preclinical value, especially in terms of immunotherapeutic approaches. Humanised mice for GC-PDXs have been described previously [101]. Mice that were pre-infused with human peripheral blood mononuclear cells (PBMCs) were implanted subcutaneously with gastric tumour tissue from the same patient. Next, the authors found that treatment with anti-hCD137 (urelumab) and anti-hPD-1 (nivolumab) significantly decreased tumour growth, affirming the clinical relevance of these humanised PDX models. To our knowledge, only one study has been published in which PDAC tissue could be successfully engrafted in immunocompetent mice. Upon tumour growth, infiltration of human-derived natural killer cells was observed, which inhibited PDAC tumour growth by specifically targeting cancer stem cells [102]. Considering that PDAC is a stroma-rich tumour with immunosuppressive features, current clinical approved immunotherapeutics have mostly failed in clinical trials for PDAC. Interestingly, transcriptomics on PDAC PDXs suggest a subgroup of PDAC patients with an immune enriched tumour, highlighting the clinical importance of further exploring humanised PDX models, also in the context of PDAC [25].

### 4.3. MiniPDX Models

Ideally, therapeutic decision making in the clinic would be based on the response of preclinical tumour models. This is not yet feasible, given the long latency before a PDX is established. To overcome this limitation, the “MiniPDX’’ was developed [103]. In this model, tumour cells were put into hollow fibre capsules before transplantation into an animal. This new technique significantly decreases engraftment time to 7 days. Recently, the feasibility of MiniPDX-guided treatment decisions was assessed in GC patients with liver metastases [104]. The authors reported that the given chemotherapy regimen, based on the response in MiniPDXs, was promising to prolong survival. Another study showed that in a herceptin-resistant GC patient, the MiniPDX method was relevant for drug screening, since it resulted in a clinical benefit for this particular case [105]. Larger clinical cohorts are needed to further confirm these findings.

## 5. Future Perspectives

The majority of novel targeted therapies that are being developed for GC and PDAC often fail during clinical evaluation, at least partially due to the fact that current models are not sufficiently reflective of the patient. Taking into account the low prevalence of targetable molecules or mutations [106], there is a great need for the development of innovative research models that are very similar to the human tumour. Multicellular PDO co-cultures and humanised PDX models are presented as next generation 3D methods to include important elements of the TME, increasing model representativeness to the in vivo situation.

Considering the high correlation between the original tumour and the first generation PDO/PDX models, these tools are not only valuable for preclinical research and disease modelling but also offer an assessment of drug sensitivity, paving the way towards personalised medicine in the clinic. Preliminary results of the HOPE trial recently tested the feasibility of generating PDO from PDAC patients within a relevant timeframe in a clinical setting. This proof-of-concept study highlighted the important potential of PDOs to display a patient-specific avatar, enabling accurate patient selection when targeted therapies are considered [107]. In parallel, PDX models have proven their utility to assess drug sensitivity profiles and to identify potentially new actionable targets [82]. To illustrate this, the recently presented “Gempred”, a predictive gene signature to gemcitabine, has been shown to be useful in a clinical setting [82]. Table 1 illustrates differences between first and next generation PDO/PDX models, reporting increased tumour representativeness by including TME in the next generation models. Since this type of modelling is a completely new area, more studies are necessary to evaluate the model’s stability, scalability and (clinical) applicability.

We expect that moving on to multicellular PDO co-cultures, humanised PDXs and complex organotypic systems will lead to the generation of close-to-human models. Such models will undoubtedly offer significant insights into tumour heterogeneity, which is of underestimated value, especially when it comes to difficult-to-treat gastrointestinal cancers such as GC and PDAC. We therefore believe that it is crucial that the cancer research field prioritises the firm establishment of these next generation models and implementation in (all) cancer research. Although there are promising aspects, we have to account for some important considerations associated with the development and use of proposed advanced 3D models. Collecting different tissue types and cell populations from the same patient demands an intensive collaboration between research labs and clinicians. Moreover, ethical concerns are increasing because these models are still based on non-human elements (mice and Matrigel), hampering clinical translation. To conclude, current developments and findings are encouraging, suggesting that they will eventually generate superior close-to-human preclinical models, allowing complete understanding of tumour biology and thereby facilitating and accelerating drug discovery and development in oncology.

## Figures and Tables

**Figure 1 ijms-23-03147-f001:**
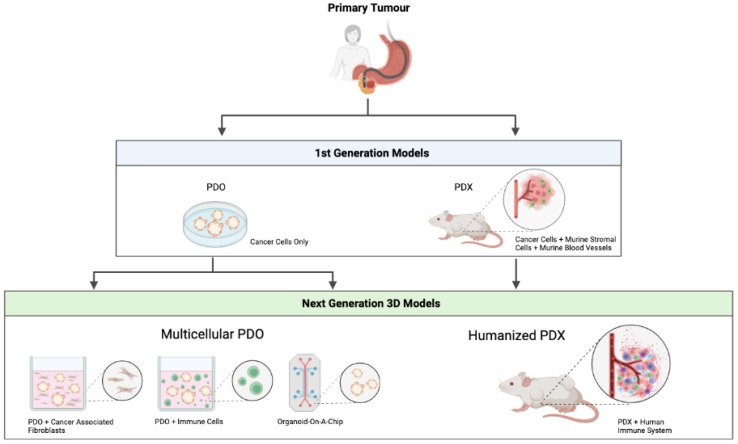
Overview of first and next generation patient-derived models. PDO—patient-derived organoid. PDX—patient-derived xenograft. Figure created with BioRender.com.

**Table 1 ijms-23-03147-t001:** Overview of the features of the various patient-derived models.

Features	PDO	PDX	Multicellular PDO	Humanised PDX
Patient tumour recapitulation	+	+	++(+) *	++
Presence of TME	−	+	−/+	++
Model stability	++	++	?	?
Establishment time	++	−	++	−
Scalability	++	+	?	−
Costs	low	high	low	very high

* Upon combination of multiple cell types in one model. PDO—patient-derived organoid. PDX—patient-derived xenograft. TME—Tumour microenvironment

## Data Availability

Not applicable.

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
