# Peer review of "Multicellular Modelling of Difficult-to-Treat Gastrointestinal Cancers: Current Possibilities and Challenges"

_ijms, 2022, doi:10.3390/ijms23063147_

Round 1

Reviewer 1 Report

The review carried out by Hakuno et al. it is very well structured, the writing follows a consistent and clear line of reasoning. Of course this topic discussed has scientific importance for the development and application of 3D multicellular modeling in the cancer treatment. In my opinion, the article is ready to be accepted for publication.

Reviewer 2 Report

The review manuscript by Sarah K. Hakuno and colleagues presents the newest close-to-human research models on gastric and pancreatic cancer. Without a doubt, there is a need to scientifically document these new generation approaches and update the know-how in order to evaluate its potential to develop new cancer immunotherapies. After a careful reading of the this review, in my opinion, the content of manuscript will be of the interest of the readers of the International Journal of Molecular Sciences. Therefore, and since I was not able to detect any major flaw, my recommendation is to considered this manuscript for publication in its present form.

Reviewer 3 Report

Dear Authors,

I enjoyed reading your manuscript (“Multicellular modelling of difficult-to-treat gastrointestinal cancers: current possibilities and challenges“, ID: ijms-1636533) and strongly recommend publishing it. I found only several minor points that could be addressed:

  1. Line 8: The “InnoSer Belgium NV“ address format does not correspond with other institutions.
  2. Line 9, 10, and 11: delete extra spaces between upper cases and addresses.
  3. Chapter 2.1: well, this is quite difficult to explain, but the chapter structure is slightly confusing (at least for me. And I understand it might be a subjective point of view. But…). First, you describe cancer (lines 62-66, this is OK). Then, there is the therapy (lines 67-76). But I do not understand why you went back to “cancer description (histological, molecular)“. It only raises a question on how this relates to the therapy (or possibly prognosis), which was not further explained (by contrast, you did a perfect job in the next chapter in this regard).
  4. Line 105: it seems that there is an extra space between “[30].“ and “To summarise“.
  5. Line 119: it seems that there is an extra space between “models“ and “significantly“.
  6. Line 139: I believe that the explanation of the asterisk should be underneath the table.
  7. Line 196: I recommend changing “subcutaneous“ to “subcutaneously“.
  8. Line 207: I recommend changing “95%-100%“ to “95–100%“.
  9. Line 273: it seems that there is an extra space between “models.“ and “ These“.
  10. Line 311: I recommend changing “50 and 70%“ to “50% and 70%“.

Best regards
